# Maximum Mean Discrepancy on Exponential Windows for Online Change Detection

**Florian Kalinke**                                                      *florian.kalinke@kit.edu*
*Karlsruhe Institute of Technology, Germany*

**Marco Heyden**                                                      *marco.heyden@kit.edu*
*Karlsruhe Institute of Technology, Germany*

**Georg Gntuni**                                                      *g.gntuni@gmail.com*
*Karlsruhe Institute of Technology, Germany*

**Edouard Fouché**                                                      *edouard.fouche@kit.edu*
*Karlsruhe Institute of Technology, Germany*

**Klemens Böhm**                                                      *klemens.boehm@kit.edu*
*Karlsruhe Institute of Technology, Germany*

**Reviewed on OpenReview:** *https://openreview.net/forum?id=OGaTF9iOxi*

## Abstract

Detecting changes is of fundamental importance when analyzing data streams and has many applications, e.g., in predictive maintenance, fraud detection, or medicine. A principled approach to detect changes is to compare the distributions of observations within the stream to each other via hypothesis testing. Maximum mean discrepancy (MMD), a (semi-)metric on the space of probability distributions, provides powerful non-parametric two-sample tests on kernel-enriched domains. In particular, MMD is able to detect any disparity between distributions under mild conditions. However, classical MMD estimators suffer from a quadratic runtime complexity, which renders their direct use for change detection in data streams impractical. In this article, we propose a new change detection algorithm, called Maximum Mean Discrepancy on Exponential Windows (MMDEW), that combines the benefits of MMD with an efficient computation based on exponential windows. We prove that MMDEW enjoys polylogarithmic runtime and logarithmic memory complexity and show empirically that it outperforms the state of the art on benchmark data streams.

## 1 Introduction

Data streams are possibly infinite sequences of observations that arrive over time. They can have different sources: sensors in industrial settings, online transactions from financial institutions, click monitoring on websites, online feeds, etc. Quickly detecting when a change takes place can yield useful insights, for example, about machine failure, malicious financial transactions, changes in customer preferences, and public opinions.

A *change* occurs if the underlying distribution of the data stream changes at a certain point in time. We call this moment *change point* (Gama, 2010); it is sometimes also referred to as *concept drift*. A principled and widely-used approach to detect changes is to use two-sample tests. The null hypothesis of such tests is that the data before and after the potential change point follow the same distribution. If the test rejects the hypothesis, one assumes that a change occurred.

One way to construct these tests is to use the kernel-based maximum mean discrepancy (MMD; Smola et al. 2007; Gretton et al. 2012), which one can interpret as a (semi-)metric on the space of probability

distributions.[1] In the statistics literature, MMD is also known as energy distance (Székely & Rizzo, 2004; 2005); see Sejdinovic et al. (2013) for the equivalence. MMD relies on the kernel mean embedding (Berlinet & Thomas-Agnan, 2004, Ch. 4); it uses a kernel function to map a probability distribution to a reproducing kernel Hilbert space (RKHS; Aronszajn 1950) and quantifies the discrepancy of the two distributions as their distance in the RKHS. MMD is a metric if the kernel mean embedding is injective; the kernel is then called characteristic (Fukumizu et al., 2008; Sriperumbudur et al., 2010). When using a characteristic kernel, the MMD two-sample test allows to distinguish any distributions given that their kernel mean embeddings exist, which is guaranteed under mild conditions.

Two-sample tests based on MMD are widely applicable, as there exist kernel functions for a multitude of Euclidean and non-Euclidean domains, for example, strings (Watkins, 1999; Cuturi & Vert, 2005), graphs (Gärtner et al., 2003; Borgwardt et al., 2020), or time series (Cuturi, 2011; Király & Oberhauser, 2019). Another benefit of kernel-based two-sample tests is their high power. While, for Euclidean data, it has been shown that the power of such tests generally decreases in the high-dimensional setting (Ramdas et al., 2015), recent results (Cheng & Xie, 2024) establish that the power rather depends on the intrinsic dimensionality of the data. The intrinsic dimensionality is typically low in real-world settings so that the kernel-based two-sample tests there do not suffer the curse of dimensionality.

Despite these benefits, a well-known bottleneck of MMD-based approaches is their computational complexity. When comparing the distributions of two sets of data of sizes $m$ and $n$, respectively, the computation of MMD with classical estimators is in $\mathcal{O}\left(m^2 + n^2\right)$, with a memory complexity in $\mathcal{O}\left(m + n\right)$. Naively computing MMD for each possible change point on a data stream with $t = m + n$ observations has a complexity in $\mathcal{O}\left(t^3\right)$ for each new observation. These properties render the direct application of MMD to change detection in data streams impractical.

In this paper, we introduce Maximum Mean Discrepancy on Exponential Windows (MMDEW), a change detection algorithm for data streams that solves the above bottleneck. Specifically, our **contributions** include the following.

- Our main contribution is MMDEW, a change detector based on an efficient online approximation of MMD. When considering the entire history of $t$ observations, the proposed method has a memory requirement of $\mathcal{O}\left(\log t\right)$ and a runtime complexity of $\mathcal{O}\left(\log^2 t\right)$ for each new observation. Otherwise, the algorithm has constant runtime and memory requirements.

- To achieve these complexities, we introduce a new data structure, which allows to approximate the quadratic time MMD in an online setting. We accomplish the speedup by introducing windows that store summaries of the observations seen so far, and by storing a sample of logarithmic size of the observations per window.

- Our experiments on standard benchmark data sets show that MMDEW performs better than state-of-the-art change detectors on four out of the five tested data sets using the $F_1$-score. For the more challenging setting of short detection delays, the proposed algorithm is better on three out of six data sets.[2]

**Outline.** Section 2 summarizes related work. Section 3 introduces the definitions and Section 4 presents the proposed algorithm. We detail the experiments in Section 5. Section 6 concludes. We include illustrative proofs in the main text but defer technical proofs, additional details, and additional experiments to the appendices.

## 2   Related work

Change detection is an unsupervised task that has received and still is receiving a lot of interest. The earliest approaches, for example, Shewhart (1925); Page (1954), originated from quality control and require strong parametric assumptions on the pre and post-change distributions. More recent work in the parametric regime weakens these assumptions by allowing post-change distributions from a parametric family with an unknown

---

[1]A function is a semimetric if it is a metric but can be zero for distinct elements.

[2]Our code is available at `https://github.com/FlopsKa/mmdew-change-detector`.

parameter (Lorden, 1970; Siegmund & Venkatraman, 1995) or by allowing any post-change distribution (Sparks, 2000; Lorden & Pollak, 2005; Abbasi & Haq, 2019; Xie et al., 2023).

In our setting, both the pre and post-change distribution are assumed to be unknown, which is a challenging setting that can be tackled with non-parametric approaches. We detail the approaches most related to our proposed method in the following and refer to Wang & Xie (2024) for a recent more extensive survey on parametric and non-parametric change detection methods.

A principled approach for comparing distributions in a stream in a non-parametric fashion is to use a corresponding statistical test. ADWIN (Bifet & Gavaldà, 2007) is a classic example but it is limited to univariate data and only detects changes in mean. ADWINK (Faithfull et al., 2019) alleviates the former by running one instance of ADWIN per feature and issues a change if a predefined number of the instances agree that a change occurred. Hence, the approach can only detect changes in the means of the marginal distributions and changes in higher moments or the covariance structure can not be detected. Still, the authors find that such an ensemble of univariate change detectors often outperforms multivariate detectors. WATCH (Faber et al., 2021) is a recent approach that uses a two-sample test based on the Wasserstein distance. However, the estimation of the Wasserstein distance requires density estimation, which is difficult for high-dimensional data (Scott, 1991). The method Dasu et al. (2009) is conceptually similar to our method, as it also relies on two-sample tests and is non-parametric, but it also requires density estimation.

In contrast, the computation of MMD-based two-sample tests does not become more difficult on high-dimensional data, which renders their usage for change detection on such data promising. We refer to Muandet et al. (2017) for a general overview of kernel mean embeddings and MMD.

There exist methods to compute MMD in the streaming setting, for example, linear time tests (Gretton et al., 2012), but their statistical power is low. Zaremba et al. (2013) introduce $B$-tests, which have higher power. However, both can not directly be used for change detection. Li et al. (2019) enable the estimation of MMD on data streams for change detection by introducing Scan $B$-statistics. Wei & Xie (2022) extend upon their work by considering multiple Scan $B$-statistics in parallel and introduce online kernel CUSUM. Another method enabling the computation of MMD on data streams is NEWMA (Keriven et al., 2020), which is based on random Fourier features (Rahimi & Recht, 2007; Sriperumbudur & Szabó, 2015), a well-known kernel approximation. NEWMA also allows detecting changes on streaming data. Harchaoui & Cappé (2007) apply kernel-based tests for offline change point detection on audio and brain-computer-interface data.

A conceptually different approach to find changes is using classifiers. D3 (Gözüaçik et al., 2019) maintains two consecutive sliding windows and trains a classifier to distinguish their elements. It reports a change if the classifier performance, measured by AUC, drops below a threshold. Another recent algorithm is IBDD (de Souza et al., 2021), which scales well with the number of features.

In our experiments in the main text, we compare MMDEW to ADWINK, WATCH, Scan $B$-Statistics, NEWMA, D3, and IBDD as these allow change detection on multivariate streams (in $\mathbb{R}^d$). These algorithms differ w.r.t. their runtime complexity, their theoretical properties, the data types that they can handle, and the types of changes that they can detect. We summarize their main properties in Table 1.[3] We consider the dimensionality $d$ as constant for the complexities where its influence is dominated by other terms and for approaches not restricted to Euclidean domains. In Appendix D.1 and Appendix D.2, we collect additional experiments on synthetic data. In particular, we additionally compare the proposed method to online kernel CUSUM and to multivariate adaptations of the Cramer-von-Mises change point model (CvM CPM; Ross & Adams 2012) and non-parametric Focus (Romano et al., 2023).

## 3 Definitions and background

This section defines our problem and recalls kernels, the mean embedding, maximum mean discrepancy, and two-sample testing.

---

[3][a]We refer to their used implementation of the Wasserstein distance computation and the discussion therein (Mérigot, 2011, Ch. 6). [b]$m$ is the number of random Fourier features and $m \ll d$. [c]The complexity results from the matrix inversion of the logistic regression model, which has cubic runtime cost. [d]Size of the constructed $q \times p$ image.

Table 1: Comparison of change detectors. Complexity — runtime complexity per new observation, ARL / MTD — type of known results, domain — data types, $t$ — total number of observations, $d$ — dimensionality (for Euclidean spaces), $k$ — parameter, $W$ — window length / block size, $N$ — number of windows.

| Algorithm | Complexity | ARL / MTD | Domain |
|---|---|---|---|
| ADWINK | $\mathcal{O}\left(dk \log W\right)$ | empirical | $\mathbb{R}^d$ |
| WATCH | unknown[a] | empirical | $\mathbb{R}^d$ |
| Scan $B$ | $\mathcal{O}\left(NW^2\right)$ | analytical | topological |
| NEWMA | $\mathcal{O}\left(md\right)$[b] | analytical | $\mathbb{R}^d$ |
| D3 | $\mathcal{O}\left(W^3\right)$[c] | none | $\mathbb{R}^d$ |
| IBDD | $\mathcal{O}\left(pq\right)$[d] | none | $\mathbb{R}^d$ |
| **MMDEW** | $\mathcal{O}\left(\log^2 t\right)$ | empirical | topological |

**Problem definition.** Let $(\mathcal{X}, \tau_{\mathcal{X}})$ be a topological space, $\mathcal{B}(\tau_{\mathcal{X}})$ the Borel sigma-algebra induced by $\tau_{\mathcal{X}}$, and $\mathcal{M}_1^+(\mathcal{X})$ the set of probability measures on $\mathcal{X}$ meant w.r.t. the measurable space $(\mathcal{X}, \mathcal{B}(\tau_{\mathcal{X}}))$. We consider a data stream, that is, a possibly infinite sequence of observations, $x_1, x_2, \ldots, x_t, \ldots$ for $t = 1, 2, \ldots$, and $x_t \in \mathcal{X}$. Each $x_t$ is generated independently following some distribution $D_t \in \mathcal{M}_1^+(\mathcal{X})$. If there exists $t^*$ such that for $i < t^*$ and $j \geq t^*$ we have $D_i \neq D_j$, then $t^*$ is a change point, and our task is to detect it; in practice, a $D_t$ typically generates a range of i.i.d. observations. We note that these definitions place few assumptions on the type of data, that is, we only require the data to reside in a topological space.

**Kernel mean embedding.** Let $\mathcal{H}$ be a reproducing kernel Hilbert space (RKHS) on $\mathcal{X}$, which means that the linear evaluation functional $\delta_x : \mathcal{H} \to \mathbb{R}$ defined by $\delta_x(f) = f(x)$ is bounded for all $x \in \mathcal{X}$ and $f \in \mathcal{H}$. By the Riesz representation theorem (Reed & Simon, 1972), there exists for each $x \in \mathcal{X}$ a unique vector $\phi(x) \in \mathcal{H}$ such that for every $f \in \mathcal{H}$ it holds that $f(x) = \delta_x(f) = \langle f, \phi(x) \rangle$. The function $\phi(x)$ is the reproducing kernel for $x$ and also called feature map; it has the canonical form $x \mapsto k(\cdot, x)$, with the function $k : \mathcal{X} \times \mathcal{X} \to \mathbb{R}$ the reproducing kernel associated to $\mathcal{H}$. With this kernel, it holds that $k(x_1, x_2) = \langle \phi(x_1), \phi(x_2) \rangle = \langle k(\cdot, x_1), k(\cdot, x_2) \rangle$ for all $x_1, x_2 \in \mathcal{X}$ (Steinwart & Christmann, 2008). The mean embedding of a probability measure $\mathbb{P} \in \mathcal{M}_1^+(\mathcal{X})$ is the element $\mu(\mathbb{P}) \in \mathcal{H}$ such that $\mathbb{E}_{X \sim \mathbb{P}}[f(X)] = \langle f, \mu(\mathbb{P}) \rangle$ for all $f \in \mathcal{H}$. The mean embedding $\mu(\mathbb{P})$ exists if $k$ is measurable and bounded (Sriperumbudur et al., 2010, Prop. 2), which we assume throughout the article.

**Maximum mean discrepancy.** MMD is defined by $\mathrm{MMD}(\mathbb{P}, \mathbb{Q}) = \|\mu(\mathbb{P}) - \mu(\mathbb{Q})\|$, where $\mu(\mathbb{P}), \mu(\mathbb{Q}) \in \mathcal{H}$ are the mean embeddings of $\mathbb{P}, \mathbb{Q} \in \mathcal{M}_1^+(\mathcal{X})$, respectively.

Let $X \sim \mathbb{P}$, $Y \sim \mathbb{Q}$ and $X'$, $Y'$ independent copies of $X$, $Y$, respectively. The squared population MMD (Gretton et al., 2012, Lemma 6) then takes the form

$$\mathrm{MMD}^2(\mathbb{P}, \mathbb{Q}) = \mathbb{E}\left[k(X, X')\right] + \mathbb{E}\left[k(Y, Y')\right] - 2\mathbb{E}\left[k(X, Y)\right],$$

where the expectations are taken w.r.t. to all sources of randomness. For observations $\hat{\mathbb{P}}_m = \{x_1, \ldots, x_m\} \overset{\text{i.i.d.}}{\sim} \mathbb{P}$ and $\hat{\mathbb{Q}}_n = \{y_1, \ldots, y_n\} \overset{\text{i.i.d.}}{\sim} \mathbb{Q}$, a biased estimator is obtained by replacing the population means with their empirical counterparts

$$\mathrm{MMD}^2\left(\hat{\mathbb{P}}_m, \hat{\mathbb{Q}}_n\right) = \frac{1}{m^2} \sum_{i,j=1}^{m} k(x_i, x_j) + \frac{1}{n^2} \sum_{i,j=1}^{n} k(y_i, y_j) - \frac{2}{mn} \sum_{i,j=1}^{m,n} k(x_i, y_j). \tag{1}$$

The runtime complexity of (1) is in $\mathcal{O}\left(m^2 + n^2\right)$. We will base our proposed approximation on (1).

**Two-sample testing.** To decide whether the value of $\mathrm{MMD}\left(\hat{\mathbb{P}}_m, \hat{\mathbb{Q}}_n\right)$ indicates a significant difference between $\mathbb{P}$ and $\mathbb{Q}$, one tests the null hypothesis $H_0 : \mathbb{P} = \mathbb{Q}$ versus its alternative $H_1 : \mathbb{P} \neq \mathbb{Q}$ by defining an acceptance region for a given level $\alpha \in (0, 1)$, which takes the form $\mathrm{MMD}\left(\hat{\mathbb{P}}_m, \hat{\mathbb{Q}}_n\right) < \epsilon_\alpha$. One rejects $H_0$ if the test statistic exceeds the threshold. The level $\alpha$ is a bound for the probability that the tests rejects $H_0$ incorrectly (Casella & Berger, 1990). Assuming that $k$ is nonnegative and bounded by $K > 0$, that is,

$0 \leq k(x, y) \leq K$ for all $x, y \in \mathcal{X}$, Gretton et al. (2012, Corollary 9) provides the distribution-free threshold $\epsilon_\alpha$ for the case that both samples $\hat{\mathbb{P}}_m$ and $\hat{\mathbb{Q}}_m$ have the same size ($m = n$) as

$$\text{MMD}\left(\hat{\mathbb{P}}_m, \hat{\mathbb{Q}}_m\right) < \sqrt{\frac{2K}{m}}\left(1 + \sqrt{2\log\frac{1}{\alpha}}\right). \tag{2}$$

Computing (2) costs $\mathcal{O}(1)$. As the change detection setting requires the case that $m \neq n$, we extend their threshold accordingly in what follows.

## 4 Our proposed algorithm

We introduce MMDEW in three steps. We first extend the threshold for the MMD two-sample test, (2), to samples of unequal sizes (Section 4.1). We then introduce our data structure that enables the efficient computation of MMD on data streams (Section 4.2). Last, we describe the complete algorithm in Section 4.3.

### 4.1 Threshold for the hypothesis test

Given a sequence of observations $\{x_1, \ldots, x_t\}$ up until time $t$ our goal is to test the null hypothesis $\mathbb{P} = \mathbb{Q}$ for any two neighboring windows $X \cdot Y = \{x_1, \ldots, x_i\} \cdot \{x_{i+1}, \ldots, x_t\}$, with $i = 1, \ldots, t-1$. Our following proposition extends Gretton et al. (2012, Theorem 8), which considers the case $m = n$, giving the distribution-free acceptance region for $m \neq n$ (corresponding to the setting that one generally encounters in change detection). The proof is deferred to Appendix A.1.

**Proposition 1.** *Let* $\mathbb{P}, \mathbb{Q} \in \mathcal{M}_1^+(\mathcal{X})$, $\hat{\mathbb{P}}_m = \{x_1, \ldots, x_m\} \overset{i.i.d.}{\sim} \mathbb{P}$, $\hat{\mathbb{Q}}_n = \{y_1, \ldots, y_n\} \overset{i.i.d.}{\sim} \mathbb{Q}$. *Assume that* $0 \leq k(x, y) \leq K$ *for all* $x, y \in \mathcal{X}$ *and* $t > 0$. *Then a hypothesis test of level at most* $\alpha > 0$ *for* $\mathbb{P} = \mathbb{Q}$ *has the acceptance region*

$$\text{MMD}\left(\hat{\mathbb{P}}_m, \hat{\mathbb{Q}}_n\right) < \sqrt{\frac{K}{m} + \frac{K}{n}}\left(1 + \sqrt{2\log\alpha^{-1}}\right) =: \epsilon_\alpha.$$

Note that, when considering multiple possible change points, one needs to account for multiple testing in order to achieve an overall level of size $\alpha$. For example, one may adjust $\epsilon_\alpha$ through Bonferroni correction ($\epsilon'_\alpha = \epsilon_\alpha/(t-1)$) by dividing by the total number of tests.

To perform change detection with MMD, it is natural to consider the stopping time

$$T = \inf\left\{t : \max_{n=1,\ldots,t-1}[\text{MMD}(\mathbb{P}_m, \mathbb{Q}_n) \geq \epsilon_\alpha] = 1\right\}, \tag{3}$$

for $m = t - n$, the empirical measures $\mathbb{P}_m = \{x_1, \ldots, x_m\}$, $\mathbb{Q}_n = \{x_{m+1}, \ldots, x_t\}$, and the brackets equal to one if their argument is true and zero otherwise (Graham et al., 1994); we note that $\epsilon_\alpha := \epsilon_\alpha(m, n)$ depends on the respective sizes of the subsamples considered. In other words, a change is indicated by the first time any MMD estimated across all splits exceeds its threshold. However, due to the quadratic runtime requirements of MMD, the computation of (3) costs $\mathcal{O}(t^3)$ for each new observation.

We now introduce our novel data structure that allows considering multiple possible change points efficiently.

### 4.2 Proposed data structure

One common method to obtain a good runtime complexity in change detection algorithms is to slice the data into windows of exponentially increasing sizes (Bifet & Gavaldà, 2007). Recent observations are collected in smaller windows, and older observations are grouped into larger windows. This leads to a fine-grained change detection in the recent past and more coarse-grained change detection in the distant past.

Our new data structure adopts this concept and, at the same time, facilitates the computation of MMD. In what follows, we first describe the properties of the proposed data structure. Then, we show how to update the data structure and explain its use for change detection.

### 4.2.1 Properties

We use 2 as the basis for the exponential slicing. Then, after observing $t$ elements, the number of windows stored in the data structure corresponds to the number of ones in the binary representation of $t$. We may thus index the windows as $B_l, \ldots, B_0$ (in decreasing order), with the largest position being $l = \lfloor \log_2 t \rfloor$. A window does not exist if the binary representation of $t$ at this position is zero.

If it exists, a window $B_s = (\mathrm{X}_s, \mathrm{XX}_s, \mathrm{XY}_s)$ at position $s = 0, \ldots, l$ stores $2^s$ observations

$$\mathrm{X}_s = \left\{ x_1^{(s)}, \ldots, x_{2^s}^{(s)} \right\}, \tag{4}$$

together with the summaries

$$\mathrm{XX}_s = \sum_{i,j=1}^{2^s} k\left( x_i^{(s)}, x_j^{(s)} \right), \tag{5}$$

$$\mathrm{XY}_s = \left\{ \underbrace{\sum_{i=1}^{2^s} \sum_{j=1}^{2^{s+1}} k\left( x_i^{(s)}, x_j^{(s+1)} \right)}_{=: \mathrm{XY}_s^{s+1}}, \ldots, \underbrace{\sum_{i=1}^{2^s} \sum_{j=1}^{2^l} k\left( x_i^{(s)}, x_j^{(l)} \right)}_{=: \mathrm{XY}_s^l} \right\}, \tag{6}$$

where $\mathrm{XX}_s \in \mathbb{R}$ is the sum of the kernel $k$ evaluated on all pairs of the window's own observations, and $\mathrm{XY}_s$ stores a list of sums of the kernel evaluated on the window's own observations and the observations in windows coming before it.[4] Storing a list enables the efficient merging of windows, elaborated in Lemma 2. The length of the list $\mathrm{XY}_s$ equals the number of windows having observations older than window $B_s$ and is at most $\lfloor \log_2 t \rfloor$. We use $\mathrm{XY}_i^j$ to represent the entry in $\mathrm{XY}_i$ that refers to the window $B_j$. Specifically, in (6), $\mathrm{XY}_s^{s+1}$ stores the interaction of $B_s$ with $B_{s+1}$; similarly, $\mathrm{XY}_s^l$ stores its interaction with $B_l$.

**Remark 1.** *Given a stream of data $x_1, x_2, \ldots, x_t$, (4) corresponds to the mapping $x_i^{(s)} = x_\ell$, with $\ell = \sum_{j=s+1}^{\lfloor \log t \rfloor} 2^j [B_j \text{ exists}] + i$, where the bracket is one if the argument is true and zero otherwise (using Iverson's convention; Graham et al. 1994). A bucket $B_j$ exists if the $j$-th right-most digit in the binary expansion of $t$ is 1.*

We summarize two of the main properties of the data structure as lemmas. Lemma 1 establishes that one can compute the value of MMD between two windows with constant complexity. The proof follows from comparing (5) and (6) with (1). Lemma 2 shows that windows can be merged with logarithmic runtime complexity. These results provide our first steps towards efficiently computing MMD in a data stream.

**Lemma 1.** *Let $B_{s+1}$ and $B_s$ be any two neighboring windows with elements $\mathrm{X}_{s+1} = \left\{ x_1^{(s+1)}, \ldots, x_{2^{s+1}}^{(s+1)} \right\}$ and $\mathrm{X}_s = \left\{ x_1^{(s)}, \ldots, x_{2^s}^{(s)} \right\}$, and sums as defined by (5) and (6), respectively. Then*

$$\mathrm{MMD}^2\left( \mathrm{X}_{s+1}, \mathrm{X}_s \right) = \frac{1}{(2^{s+1})^2} \mathrm{XX}_{s+1} + \frac{1}{(2^s)^2} \mathrm{XX}_s - \frac{2}{(2^{s+1})(2^s)} \mathrm{XY}_s^{s+1},$$

*with a computational complexity of $\mathcal{O}(1)$.*

**Lemma 2.** *Merging two windows $B_{s+1}$ and $B_s$ into a new window $B'$, such that $B'$ stores (4), (5), and (6) costs $\mathcal{O}(\log t)$.*

Besides showing the result, the proof of Lemma 2 illustrates the steps that allow merging windows efficiently.

---

[4]Note that the superscript $(s)$ of the $x_i^{(s)}$-s indicates the corresponding window $B_s$.

Figure 1: Schematic representation of Example 1. For a given step, the proposed scheme stores the windows in bold face.

*Proof.* For computing $XX'$, we use the symmetry of $k$ to obtain

$$
XX' = \sum_{i,j=1}^{2^{s+1}} k\left(x_i^{(s+1)}, x_j^{(s+1)}\right) + \sum_{i,j=1}^{2^s} k\left(x_i^{(s)}, x_j^{(s)}\right) + \sum_{i=1}^{2^{s+1}}\sum_{j=1}^{2^s} k\left(x_i^{(s+1)}, x_j^{(s)}\right) + \sum_{i=1}^{2^s}\sum_{j=1}^{2^{s+1}} k\left(x_i^{(s)}, x_j^{(s+1)}\right)
$$

$$
= \sum_{i,j=1}^{2^{s+1}} k\left(x_i^{(s+1)}, x_j^{(s+1)}\right) + \sum_{i,j=1}^{2^s} k\left(x_i^{(s)}, x_j^{(s)}\right) + 2\sum_{i=1}^{2^{s+1}}\sum_{j=1}^{2^s} k\left(x_i^{(s+1)}, x_j^{(s)}\right) = XX_{s+1} + XX_s + 2XY_s^{s+1},
$$

(7)

which has a runtime complexity in $\mathcal{O}(1)$.

To compute $XY'$, we note that $B_{s+1}$ stores the list $XY_{s+1}$ of kernel evaluations corresponding to all windows coming before it. The same holds for $B_s$, for which the list has one more element, $XY_s^{s+1}$, which was used in (7). All the elements in $XY_s$ and $XY_{s+1}$ are sums and thus additive; it suffices to merge both lists by adding their values element-wise, omitting $XY_s^{s+1}$, and storing the result in $XY'$. As each list has at most $\log t$ elements, merging them is in $\mathcal{O}(\log t)$. $\qquad\square$

Specifically, the scheme facilitates the merging of windows of equal size, enabling us to establish the exponential structure outlined in the next section.

### 4.2.2 Insertion of observations

The structure is set up recursively. For each new observation, we create a new window $B_0$, with $XX_0$ as defined by (5) and $XY_0$ computed w.r.t. the already existing windows. If two windows have the same size, we merge them by Lemma 2, which costs $\mathcal{O}(\log t)$. This yields $\lfloor \log t \rfloor$ windows of exponentially increasing sizes.

We illustrate the scheme in the following Example 1 and the corresponding Figure 1.

**Example 1.** *To set up the structure, we start with the first observation $x_1$ and create the first window $B_0$, with $XX_0$ as defined by (5) and $XY_0 = \emptyset$. When observing $x_2$, we similarly create a new window $B_0'$, now also computing $XY_{0'}^0 = \{XY_{0'}^0\}$. As $B_0$ and $B_0'$ have the same size, we merge them into $B_1$, computing $XX_1$ with (7). No previous window exists so that $XY_1 = \emptyset$. We repeat this for all new observations, for example, for $x_3$, one creates (a new) $B_0$, computing $XX_0$ and $XY_0 = \{XY_0^1\}$, which results in two windows, $B_1$ and $B_0$.*

### 4.2.3 MMD computation and change detection

We now show that we can compute the MMD statistic (1) at positions between windows with a runtime complexity of $\mathcal{O}(\log t)$.

**Proposition 2.** *Let $B_l, \ldots, B_{s+1}, B_s, \ldots, B_0$ be a given list of windows with corresponding elements $\mathrm{X}_i$, $i = 0, \ldots, l$, as defined in (4). For any split $s \in \{1, \ldots, l-1\}$, the computation of*

$$\mathrm{MMD}^2 \left( \bigcup_{i=s+1}^{l} \mathrm{X}_i, \bigcup_{i=0}^{s} \mathrm{X}_i \right), \tag{8}$$

*that is, the computation of MMD between the elements in windows coming before window $B_s$ and the elements in windows coming after (and including) $B_s$, has a runtime complexity of $\mathcal{O}\left(\log t\right)$ for $0 < s < l$, with $s, l \in \mathbb{N}$.*

*Proof.* To obtain (8), one recursively merges $B_s, \ldots, B_0$ to $B'_s$ using Lemma 2, starting from the right, and similarly $B_l, \ldots, B_{s+1}$ to $B'_l$. One then obtains the statistic with Lemma 1, and by setting $\mathrm{XY}^{l'}_{s'} = \sum_{i=1}^{l-s} \mathrm{XY}^i_{s'}$, that is, by summing all elements in the $\mathrm{XY}'_s$-list of $B'_s$. This concludes the proof as the logarithmic complexity was already established. □

The application of the presented data structure for change detection is as follows. For each new observation, we estimate MMD at any position between windows and compare it to the threshold $\epsilon'_\alpha = \frac{\epsilon_\alpha}{l}$ (with Bonferroni correction) from Proposition 1. We report a change when the value of MMD exceeds the threshold. As there are at most $\log t$ windows, we have at most $\log t - 1$ positions. Computing MMD for a position is in $\mathcal{O}\left(\log t\right)$ by Proposition 2, and so the procedure has a total runtime complexity of $\mathcal{O}\left(\log^2 t + t\right)$ per insert operation, where the term linear in $t$ results from computing $\mathrm{XY}_0$ when inserting a new observation. We may equivalently consider the proposed method as providing a more coarse-grained estimate of (3), taking the form

$$T' = \inf \left\{ t : \max_{s=1,\ldots,l} \left[ \mathrm{MMD} \left( \bigcup_{i=s+1}^{l} \mathrm{X}_i, \bigcup_{i=0}^{s} \mathrm{X}_i \right) \geq \epsilon_\alpha \right] = 1 \right\}, \tag{9}$$

with the $X_i$-s defined as in Proposition 2.[5]

While the data structure in its current form allows to obtain the precise values of (1) in an incremental fashion, its runtime and memory complexity are $\mathcal{O}\left(t\right)$ for each new observation; these complexities are unsuitable for deploying the algorithm in the streaming setting. We reduce the runtime by subsampling within the windows, which we present together with the complete algorithm in the following section.

## 4.3 MMDEW Algorithm

Our algorithm builds upon the data structure discussed previously. But, we suggest that each window of size $2^s$, $s = 0, \ldots, l$, samples $s$ observations (of the total $2^s$), that is, a logarithmic amount, while keeping everything else as before.

In this section, we first analyze such subsampling and discuss its benefits. Afterwards, we present the complete algorithm. We refer to Appendix A.2 for the proof of the following statement.

**Proposition 3.** *With subsampling, the number of terms in the sum $\mathrm{XX}_l$ for a window at position $l$, $1 \leq l$, $l \in \mathbb{N}$ is*

$$n_{\mathrm{XX}_l} = 2^{l-1} \left( l^2 - l + 4 \right) = \frac{t}{2} \left( \log_2^2 t - \log_2 t + 4 \right),$$

*with $t = 2^l$ the number of observations of $B_l$. The number of terms of $\mathrm{XY}^l_l$ for windows of the same size, which occur prior to merging, is*

$$n_{\mathrm{XY}^l_l} = 2^l l = t \log_2 t.$$

---

[5]Recall that Remark 1 gives the precise value of $l$. Further, some split positions $s$ may not exist.

**Input:** Data stream $x_1, x_2, \ldots$, level $\alpha$
**Output:** Change points in $x_1, x_2, \ldots$; detection times

1:    $windows \leftarrow \emptyset$                          ▷ List of windows
2:    **for** each $x_i \in \{x_1, x_2, \ldots\}$ **do**
3:        $X_0 \leftarrow x_i$                           ▷ Initialize $B_0$
4:        $XX_0 \leftarrow k(x_i, x_i)$
5:        **for** each $B_j \in windows$ **do**
6:           $XY_0^j \leftarrow \sum_{x_k^{(j)} \in B_j} k(x_i, x_k^{(j)})$
7:        $B_0 = (X_0, XX_0, XY_0)$
8:        $windows \leftarrow windows \cup B_0$
9:        **for** each split $s$ in $windows = \{B_l, \ldots, B_{s+1}, B_s, \ldots, B_0\})$ **do**      ▷ Detect changes
10:           **if** $MMD\left(\bigcup_{j=s+1}^{l} X_j, \bigcup_{j=0}^{s} X_j\right) \geq \epsilon'_\alpha$ **then**
11:              **print** "Change at $s$ detected at time $i$"
12:              $windows \leftarrow B_s, \ldots, B_0$              ▷ Drop windows
13:        **while** two windows have the same size $2^l$ **do**       ▷ Maintain exponential structure
14:           Merge windows following Lemma 2 into $B_{l+1}$
15:           Store a uniform sample of size $l + 1$ in $X_{l+1}$ of $B_{l+1}$

Algorithm 1: Proposed MMDEW change detection algorithm.

**Remark 2.** *The number of terms in the sums of (1) acts as a proxy for the quality of the estimate. It is optimal when no subsampling takes place; this number is $\mathcal{O}\left(t^2\right)$. When subsampling a logarithmic number of observations per window with our data structure (as we propose), one achieves polylogarithmic runtime and logarithmic memory complexity. At the same time, one achieves a better approximation quality than naively sampling a logarithmic number of observations without the summary data structure. While such sampling would also yield a memory complexity of $\mathcal{O}\left(\log t\right)$ when using the naive approach for change detection—that is, splitting the sample into two neighboring windows and computing $MMD^2$—the number of terms in (1) would be $\mathcal{O}\left(\log^2 t\right)$. Proposition 3 shows that the summary data structure improves upon this by a factor of approximately $t/2$ for $n_{XX_l}$ and a factor of $t/\log_2 t$ for $n_{XY_l^l}$ (we neglect logarithmic and constant terms in the former due to their small contribution).*

Algorithm 1 now summarizes the complete algorithm, with MMD in Line 10 referring to the computation of MMD as in Proposition 2. MMDEW stores only a uniform sample of size $l + 1$, that is, of size logarithmic in the number of observations, while keeping the respective $XX_s$ and $XY_s$, $s = 0, \ldots, l$, computed before. With this approach, the number of samples in a window increases by one each time the window is merged, and the memory complexity is logarithmic in the number of observations. Note that one recovers the previous algorithm (Section 4.2.3) and therefore the precise value of (8) if one omits Line 15. Further, changes in Line 15 allow to adjust the subsampling, for example, the user may defer the sampling until windows contain a minimum number of observations, or choose a different function to control the sample size.

The following example illustrates the procedure. Figure 2 expands upon Example 2 and shows the evolution of the data structure upon observing $x_1, \ldots, x_6$ and when merging windows.

**Example 2.** *We assume that there is a stream of i.i.d. observations $x_1, x_2, \ldots$. Note that the i.i.d. assumption implies that there are no changes. MMDEW receives the first observation, $x_1$ and creates a window $B_0$ storing $x_1$, $XX_0 = k(x_1, x_1)$, and $XY_0 = \emptyset$. For the next observation, $x_2$, it creates a new window $B_{0'}$, storing $x_2$, $XX_{0'} = k(x_2, x_2)$, and $XY_{0'} = \{k(x_1, x_2)\}$ and detects no change. As $B_0$ and $B_{0'}$ have the same size, MMDEW merges them into window $B_1$, storing a sample of size $\log_2 2 = 1$, say, it stores $x_1$ and discards $x_2$, and computes $XX_1 = k(x_1, x_1) + k(x_2, x_2) + 2k(x_1, x_2)$, following (5). As no previous window exists, the computation of $XY_1$ is not required. We see that the number of terms in $XX_1$ equals four, while $B_1$ stores only one observation (established in Proposition 3). Next, the algorithm observes $x_3$ and creates a new window, $B_0$, storing $x_3$, $XX_0 = k(x_3, x_3)$, and computing $XY_0$ to the window coming before, that is, $B_1$, so that $XY_0 = \{XY_0^1\}$. In the next step, MMDEW receives $x_4$, again creating a new window $B_{0'}$.*

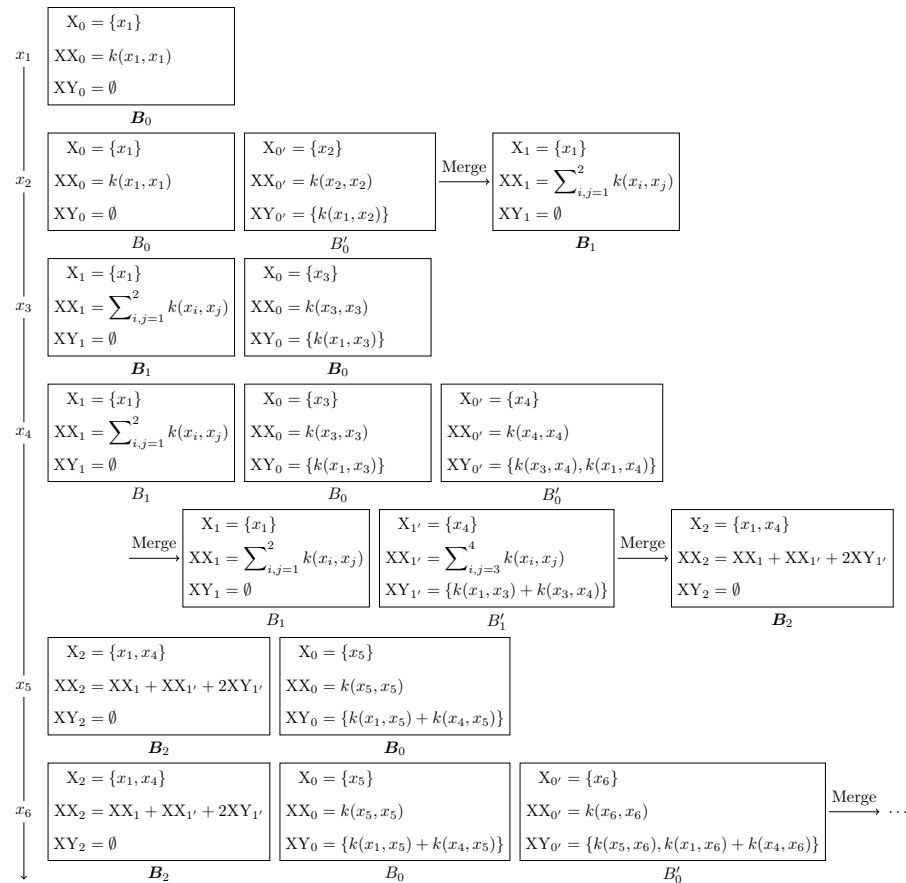

Figure 2: Set up of data structure with subsampling upon inserting $x_1, \ldots, x_6$. MMDEW stores the windows in bold face at the end of the merge operations. Observations $x_2$ and $x_3$ are not stored explicitly due to the sampling applied. $x_4$ is split into two lines for readability. See Example 2 for a detailed discussion.

*The algorithm now recursively merges the windows, that is, $B_0$ and $B_{0'}$ become $B_{1'}$, and $B_1$ and $B_{1'}$ then become $B_2$. Upon receiving $x_5$, the algorithm creates a new window $B_0$, storing $x_5$, the kernel evaluation $k(x_5, x_5)$, and the interaction of $x_5$ with $\{x_1, x_4\}$ from $B_3$. We conclude the example with $x_6$, which leads to the creation of a new window $B_0'$. As in steps 2 and 4, $B_0$ and $B_0'$ will now be merged to obtain $B_1$.*

Algorithm 1 has a runtime cost of $\mathcal{O}\left(\log^2 t\right)$ per insert operation and a total memory complexity of $\mathcal{O}\left(\log t\right)$. This allows it to scale to very large data streams. Nevertheless, if one strictly requires constant time and memory, one can simply limit the number of windows at the expense of detecting changes only up to a certain time in the past. In the latter configuration, MMDEW fulfills the requirements for streaming algorithms laid out by Domingos & Hulten (2003).

## 5    Experiments

This section showcases our approach on synthetic data (Section 5.1) and on streams derived from real-world classification tasks (Section 5.2). We ran all experiments on a server running Ubuntu 20.04 with 124GB RAM, and 32 cores with 2GHz each.

## 5.1 Synthetic data

To evaluate the average run length (ARL) and the mean time to detection (MTD) in a controlled environment, we first conduct experiments on synthetic data, comparing MMDEW to the MMD estimate (1) as baseline.[6] We also compare the runtime of MMDEW to that of existing change detectors. For an in-depth comparison of the ARL and MTD trade-off for kernel-based approaches with optimally chosen thresholds, see Appendix D.1. For a comparison of MMDEW with univariate approaches, we refer to Appendix D.2.

**ARL and MTD.** The ARL quantifies the expected number of observations processed before a change detector flags a change, assuming $H_0$ holds. In the static setting, this corresponds to the type I error. Formally, for $\tilde{T}$ corresponding to the stopping time captured by Algorithm 1, that is, (9) with subsampling applied, we are interested in $\mathbb{E}_{H_0}\tilde{T}$.

The error under the alternative ($H_1$ holds) is captured by the expected detection delay (EDD), also called "mean time to detection (MTD)". Specifically, a change detector processes a stream that contains a change at a known observation $\kappa$ ($\kappa = 1, \ldots, t$) and we want to know the delay until the change is reported $\mathbb{E}_{x_1,\ldots,x_{\kappa-1}\sim\mathbb{P},x_\kappa,\ldots,x_t\sim\mathbb{Q}}\tilde{T}$, with $\mathbb{P}$ the pre-change distribution and $\mathbb{Q} \neq \mathbb{P}$ the post-change distribution. In other words, the EDD quantifies how many samples of $\mathbb{Q}$ must be processed to flag a change after having observed $\kappa - 1$ samples from $\mathbb{P}$. Note that $\kappa$ must be chosen large enough to allow MMD to capture the difference in $\mathbb{P}$ and $\mathbb{Q}$, and we assume that the statistic does not exceed the treshold on the first $\kappa$ samples. In a static setting, the EDD is comparable with the type II error. Note that existing literature (Xie & Siegmund, 2013; Wei & Xie, 2022) usually considers a fixed threshold $b$ for the stopping time for both ARL and EDD, while our threshold $\epsilon_\alpha$ depends on the position of the split considered. Experiments for a fixed value of $b$ are deferred to Appendix D.1.

To approximate the ARL and the EDD in different scenarios, we simulate 5-dimensional data distributed according to the multivariate normal $\mathcal{N}\left(\mathbf{0},\mathbf{I}_5\right)$, the uniform $\mathcal{U}[-\mathbf{1}_5,\mathbf{1}_5]$, the Laplace $(0,\sigma\mathbf{I}_5)$, and a mixed distribution, respectively. The mixed distribution is taken to be $\mathcal{N}\left(\mathbf{0},\mathbf{I}_5\right)$ with probability 0.3 and $\mathcal{N}\left(\mathbf{0},\sigma^2\mathbf{I}_5\right)$ with probability 0.7, where $\mathbf{1}_d$ denotes a vector of $d$ ones and $\mathbf{I}_d$ is the $d$-dimensional identity matrix. We set $\sigma = 3$.

To compute ARL, we consider 10,000 observations distributed according to either the uniform, the Laplace, or the mixed distribution. Hence, the data does not contain any changes. For MTD, we first run both algorithms on 512 $(= 2^9)$ and 1024 $(= 2^{10})$ observations, respectively, leading to MMDEW summarizing the data in one window in both cases. These observations are distributed according to $\mathcal{N}\left(\mathbf{0},\mathbf{I}_5\right)$ and then followed by either the uniform, Laplace, or mixed distribution. That is, we induce a change point at $\kappa = 513$ (resp. $\kappa = 1025$), and then count the number of observations processed from the new distribution until the algorithms report a change.

Figure 3 collects the average results over 20 repetitions. The left plot shows that an increase in the level $\alpha \in (0,1)$ leads to a decrease in ARL. This is expected as the test becomes more sensitive, leading to more false positives. The baseline achieves a higher ARL but at the cost of an increased runtime. The MTD plots (center and r.h.s.) mirror the ARL observation: The MTD decreases with increasing $\alpha$. We further observe that the detection delay depends on the post-change distribution. The delay is comparably large when changing from the multivariate standard normal to the mixed distribution. This matches our intuition: the mixed distribution is relatively similar to the pre-change distribution, rendering it difficult to detect a change between them. For larger values of $\alpha$, that is, $\alpha \geq 0.2$, MMDEW performs similarly to the baseline in all cases. Comparing the MTD when the change happens after 512 observations to the MTD when the change happens after 1024 observations, the results show that more pre-change samples render the algorithms more sensitive to detecting changes, due to more samples improving the approximation of the mean embedding. Overall, the results on these synthetic streams indicate that MMDEW is (i) robust to the choice of $\alpha$ and (ii) that $\alpha$ has the expected influence on the behavior of the algorithm.

---

[6]For computational reasons, we compute MMD as described in the discussion following Proposition 1. F a fair comparison, we use the distribution-free bound of Proposition 1 for both algorithms.

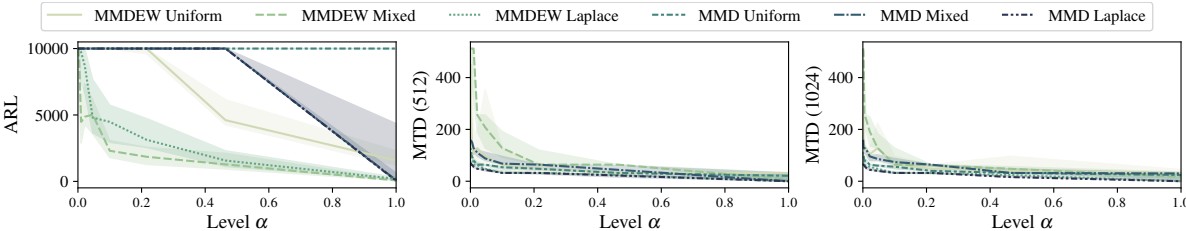

Figure 3: Average run length (ARL) and expected detection delay / mean time to detection (EDD / MTD) of MMDEW on synthetically generated data.

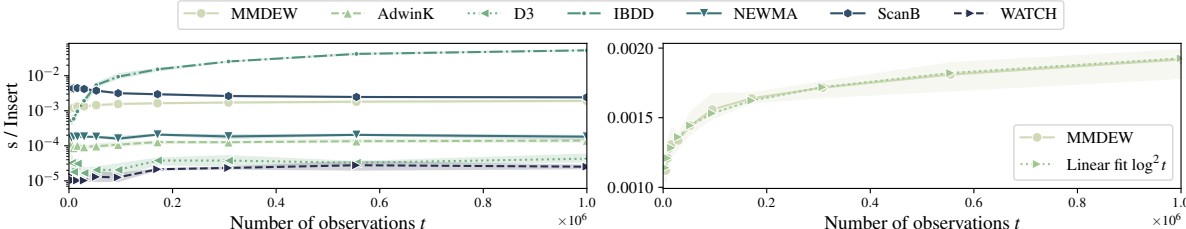

Figure 4: Comparison of runtimes per insert operation (l.h.s.) and least squares fit validating the theoretical runtime complexity of MMDEW w.r.t. the runtime observed in practice (r.h.s.).

**Runtime.** We now compare the runtime of MMDEW to that of its contenders and additionally validate the runtime guarantees that we derived analytically in Section 4.3.

To this end, we generate a constant stream of $10^6$ one-dimensional observations, that is, the observed stream contains no change. Note that, while the dimensionality of the data affects the runtime depending on the used kernel, its influence is the same across all kernel-based algorithms, hence we limit our considerations to the univariate case.

Figure 4 shows the average results over 10 runs. The left plot reveals that the fixed cost per insert of MMDEW is relatively large, as processing a small number of observations requires comparably much time. However, the runtime does not increase by much with the number of observations. The figure also shows that the proposed algorithm's runtime is better than that of an alternate kernel-based method, Scan $B$-statistics, where we use a window size of $\omega = 100$ in the runtime experiments. For $t > 0.05 \cdot 10^6$, MMDEW also outperforms IBDD. Still, the other algorithms run faster than MMDEW but achieve a lower $F_1$ score in our later experiments.

The right plot of Figure 4 verifies the analytically derived runtime of $\mathcal{O}\left(\log^2 t\right)$ by fitting the corresponding curve $(t \mapsto c \log^2 t)$ to the measured data with the least squares method. The resulting mean squared error is approximately $10^{-6}$, which confirms the preceding asymptotic runtime analysis.

## 5.2 Real-world classification data

To obtain our change detection quality estimates, we use well-known classification data sets and interpret them as streaming data.[7] This is common in the literature, for example, Faithfull et al. (2019); Faber et al. (2021), as only few high-dimensional annotated change detection data sets are publicly available.

For each data set, we first order the observations by their classes; a change occurs if the class changes. To introduce variation into the order of change points, we randomly permute the order of the classes before each run but use the same permutation across all algorithms. For preprocessing, we apply min-max scaling to all data sets. Table 2 summarizes the data sets, where $n$ is the number of observations, $d$ is the data dimensionality, and #CP is the number of change points.

---

[7]While MMDEW is not limited to Euclidean data, Euclidean data is the type of data most frequently encountered in practice, and our experiments target at this setting.

Table 2: Overview of data sets.

| Data set | $n$ | $d$ | #CPs |
|---|---|---|---|
| CIFAR10 (Krizhevsky et al., 2009) | 60,000 | 1,024 | 9 |
| FashionMNIST (Xiao et al., 2017) | 70,000 | 784 | 9 |
| Gas (Vergara et al., 2012) | 13,910 | 128 | 5 |
| HAR (Anguita et al., 2013) | 10,299 | 561 | 5 |
| MNIST (Deng, 2012) | 70,000 | 784 | 9 |

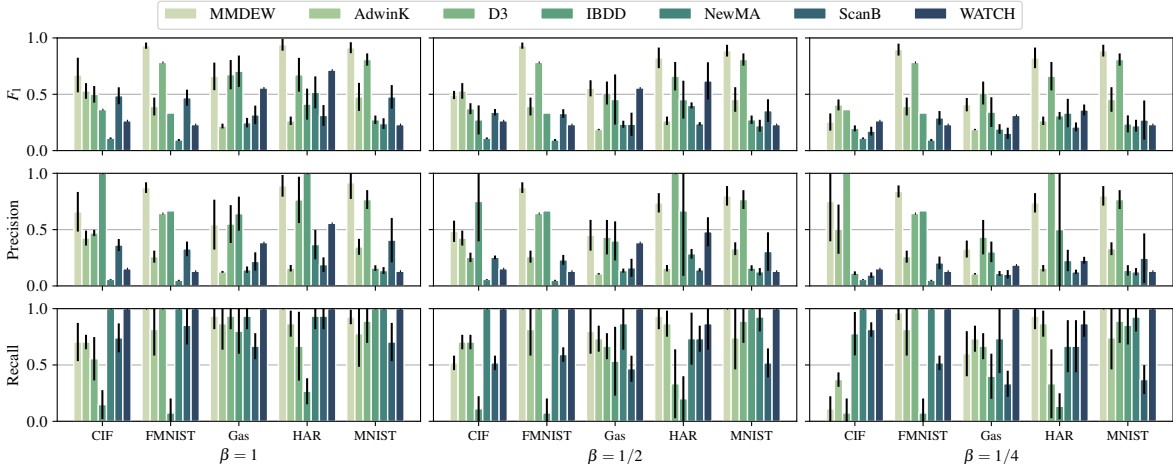

Figure 5: Average $F_1$-score, precision and recall. The bars show the standard deviation over 10 permutations of the data.

We run a grid parameter optimization per data set and algorithm and report the best result w.r.t. the $F_1$-score. We note that such an optimization is difficult to perform in practice—here one typically prefers approaches with fewer or easy-to-set parameters—but allows a fair comparison. Table 3 in Appendix C lists all the parameters we tested. We note that the grid parameter optimization allowed us to obtain better $F_1$-scores than the heuristics proposed in Keriven et al. (2020) for NEWMA and Scan $B$-statistics.

We exclude the squared time estimator of MMD due to its prohibitive runtime. For kernel-based algorithms (MMDEW, NEWMA, and Scan $B$-statistics) we use the Gaussian kernel $k(x, y) = \exp\left(-\gamma\|x - y\|^2\right)$ ($\gamma > 0$) and set $\gamma$ using the median heuristic (Garreau et al., 2018) on the first 100 observations. The Gaussian kernel is universal (Steinwart & Christmann, 2008; Szabó & Sriperumbudur, 2017) and allows, given enough data, to detect any change in distribution as a universal kernel on a compact domain is characteristic (Gretton et al., 2012, Theorem 5). We also supply the first 100 observations to competitors requiring data to estimate further parameters (IBDD, WATCH) upfront.

**$F_1$-score, precision, and recall.** We compute the precision, the recall, and the $F_1$-score, which are common to evaluate change detection algorithms (Li et al., 2019; Keriven et al., 2020; van den Burg & Williams, 2020; Faber et al., 2021). Specifically, for a fixed $\Delta_T \in \mathbb{N}_{>0}$, we proceed as follows. If a change is detected, and there is an actual change point within the $\Delta_T$ previous time steps, we consider it a true positive (tp). If a change is detected, and there is no change point within the $\Delta_T$ previous steps, we consider it a false positive (fp). If no change is detected within $\Delta_T$ steps of a change point, we consider it a false negative (fn). We count at most one true positive for each actual change point. With these definitions, the precision is Prec = tp/(tp + fp), the recall is Rec = tp/(tp + fn), and the $F_1$-score is their harmonic mean $F_1 = 2 \cdot (\text{Prec} \cdot \text{Rec}) / (\text{Prec} + \text{Rec})$. Note that, while some algorithms allow to infer where in the data a change happens, including the proposed MMDEW, we only evaluate the time at which they report a change, as all tested approaches allow reporting this value.

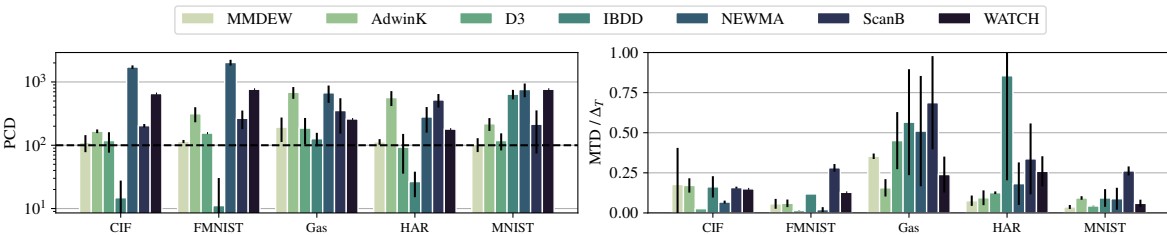

Figure 6: Average of percentage of changes detected (PCD) and of mean time to detection (MTD). The dashed line indicates the optimum for PCD. For MTD lower values are better.

Figure 5 shows our results. As $\Delta_T$ is an evaluation-specific parameter, we vary it relative to the average distance between change points by a factor $\beta > 0$: Given a data set of length $N$ with $n$ changes, we set $\Delta_T = \beta \cdot N/(n+1)$. For $\beta = 1$ ($\Delta_T$ is equal to the average number of steps between change points per respective data set), MMDEW achieves a higher $F_1$-score than all competitors on all data sets except for Gas, where it still obtains a competitive result. Throughout, the proposed algorithm obtains a good balance between precision and recall. Other approaches either have very low precision (for example, less than 20%), or an inferior recall and precision, down to a few exceptions. With a reduced $\beta$, that is, we allow only a shorter detection delay, the performance of all algorithms decreases on average. For $\beta = 1/2$, MMDEW achieves the best $F_1$ score also on four data sets, and, for $\beta = 1/4$ (the most challenging setting) on three of the tested data sets.

We conclude that the proposed method achieves very good results across all these experiments—especially when taking into account the fewer hyperparameters compared to the other approaches that we tested.

**Percentage of changes detected and detection delay.** To obtain a complete picture of the performance of MMDEW, we also report the "percentage of changes detected" (PCD), that is, the ratio of the number of reported changes and the number of actual change points, and its MTD on the data streams derived from real-world data. In our context, MTD coincides with the expected detection delay.

Figure 6 collects our results. For PCD, results closer to 100% are better. Here, MMDEW is on par with the closest competitors and consistently, that is, across all data sets, detects an approximately correct number of change points. D3, NEWMA, Scan $B$-statistics, and WATCH detect too many change points in all cases. This behavior is also reflected in their comparably large recall in Figure 5.

For MTD, lower values are better. Here, the classification-based D3 performs best in most of the cases. MMDEW performs a bit worse than D3 but better than the other algorithms on most data sets, with the Gas data set the major exception. As the experiments in Figure 5 show, a lower $\Delta_T$ tends to lead to a lower $F_1$-score of MMDEW. In other words, MMDEW tends to detect changes with some delay, but it detects them consistently.

## 6 Conclusions

We introduced a novel change detection algorithm, MMDEW, that builds upon two-sample testing with MMD, which is known to yield powerful tests on many domains. To facilitate the efficient computation of MMD, we presented a new data structure, which allows to estimate MMD with polylogarithmic runtime and logarithmic memory complexity. Our experiments on standard benchmark data show that MMDEW obtains the best $F_1$-score on most data sets. At the same time, MMDEW only has two parameters—the level of the statistical test and the choice of kernel. This simplifies the proposed algorithm's application in real-world use cases.

**Acknowledgments**

This work was supported by the German Research Foundation (DFG) Research Training Group GRK 2153: Energy Status Data—Informatics Methods for its Collection, Analysis and Exploitation and by the Baden-Württemberg Foundation via the Elite Program for Postdoctoral Researchers.

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

# A   Proofs

This section contains additional proofs. The proof of Proposition 1 is in Section A.1. Proposition 3 is proved in Section A.2.

## A.1   Proof of Proposition 1

Proposition 1 follows from the more general result that we state below. The statement and proof are similar to Gretton et al. (2012, Theorem 8) but do not assume $m = n$. Note that we recover Gretton et al. (2012, Theorem 8) in the case that $m = n$. We prove Proposition 1 afterwards.

**Proposition 4.** *Let* $\mathbb{P}$, $\mathbb{Q}$, $\hat{\mathbb{P}}_m$, $\hat{\mathbb{Q}}_n$ *be defined as in the main text, assume* $0 \leq k(x,y) \leq K$ *for all* $x, y \in \mathcal{X}$, $\mathbb{P} = \mathbb{Q}$, *and* $t > 0$. *Then*

$$P\left(\mathrm{MMD}\left(\hat{\mathbb{P}}_m, \hat{\mathbb{Q}}_n\right) - \left(\frac{K}{m} + \frac{K}{n}\right)^{\frac{1}{2}} \geq t\right) \leq e^{-\frac{t^2 mn}{2K(m+n)}}.$$

*Proof.* First, we bound the difference of $\mathrm{MMD}\left(\hat{\mathbb{P}}_m, \hat{\mathbb{Q}}_n\right)$ to its expected value. Changing a single one of either $x_i$ or $y_j$ in this function results in changes of at most $2\sqrt{K}/m$, and $2\sqrt{K}/n$, giving

$$\sum_{i=1}^{n+m} c_i^2 = 4K\frac{n+m}{nm}.$$

We now apply the bounded differences inequality (recalled in Theorem 1) to obtain

$$P\left(\mathrm{MMD}\left(\hat{\mathbb{P}}_m, \hat{\mathbb{Q}}_n\right) - \mathbb{E}\,\mathrm{MMD}\left(\hat{\mathbb{P}}_m, \hat{\mathbb{Q}}_n\right) \geq t\right) \leq e^{-\frac{t^2 mn}{2K(m+n)}}.$$

The last step is to bound the expectation, which yields

$$\mathbb{E}\,\mathrm{MMD}\left(\hat{\mathbb{P}}_m, \hat{\mathbb{Q}}_n\right) = \mathbb{E}\left(\frac{1}{m^2}\sum_{i,j=1}^{m} k(x_i, x_j) + \frac{1}{n^2}\sum_{i,j=1}^{n} k(y_i, y_j) - \frac{1}{mn}\sum_{i,j=1}^{m,n} k(x_i, y_j) - \frac{1}{mn}\sum_{j,i=1}^{n,m} k(y_j, x_i)\right)^{\frac{1}{2}}$$

$$\leq \left(\frac{1}{m}\mathbb{E}k(X,X) + \frac{1}{n}\mathbb{E}k(Y,Y) + \frac{1}{m}(m-1)\mathbb{E}k(X,Y) + \frac{1}{n}(n-1)\mathbb{E}k(Y,X) - 2\mathbb{E}k(X,Y)\right)^{\frac{1}{2}}$$

$$= \left(\frac{1}{m}\mathbb{E}k(X,X) + \frac{1}{n}\mathbb{E}k(Y,Y) - \frac{1}{m}\mathbb{E}k(X,Y) - \frac{1}{n}\mathbb{E}k(X,Y)\right)^{\frac{1}{2}}$$

$$= \left(\frac{1}{m}\mathbb{E}\left[k(X,X) - k(X,Y)\right] + \frac{1}{n}\mathbb{E}\left[k(X,X) - k(X,Y)\right]\right)^{\frac{1}{2}} \leq \left(\frac{K}{m} + \frac{K}{n}\right)^{\frac{1}{2}}.$$

Inserting this into the previous inequality, we obtain the stated result.  □

Proposition 1 is now a corollary of Proposition 4, which follows by setting $\alpha = e^{-\frac{t^2 mn}{2K(m+n)}}$ and solving for $t$ to obtain a test of level $\alpha$.

## A.2 Proof of Proposition 3

To find $n_{\mathrm{XY}_l^l}$, we use our implementation of MMDEW and the On-Line Encyclopedia of Integer Sequences (OEIS) to discover that $n_{\mathrm{XY}_l^l}$ follows the sequence $1, 2, 8, 24, 64, 160, \ldots$ for $l = 0, 1, 2, \ldots$. Thus

$$n_{\mathrm{XY}_l^l} = 2^l l, \quad \text{for } l > 0 \tag{10}$$

and $n_{\mathrm{XY}_0^0} = 1$ (Sloane, 1999b).

To find $n_{\mathrm{XX}_l}$, notice that $n_{\mathrm{XX}_l}$ only changes when one merges two windows, which happens for windows of the same size $n_{\mathrm{XX}_{l-1}}$. The algorithm adds to this $2 \cdot n_{\mathrm{XY}_{l-1}^{l-1}}$ terms, see (7), and, for $l = 0, 1, 2, \ldots$, we obtain the recurrence relation

$$n_{\mathrm{XX}_l} = \begin{cases} 1 & \text{if } l = 0, \\ 4 & \text{if } l = 1, \\ 2 \cdot n_{\mathrm{XX}_{l-1}} + 2 \cdot n_{\mathrm{XY}_{l-1}^{l-1}} & \text{if } l > 1, \end{cases}$$

with $n_{\mathrm{XX}_{-1}} := 0$. Now write

$$n_{\mathrm{XX}_l} = 2 \cdot n_{\mathrm{XX}_{l-1}} + l \cdot 2^l - 2^l + 2 \cdot [l = 0] + 2 \cdot [l = 1], \tag{11}$$

where the brackets are equal to one if their argument is true and zero otherwise (using Iverson's convention; Graham et al. 1994). To find a closed-form expression for (11), we define the ordinary generating function $A(z) = \sum_l a_l z^l$. Now, we multiply (11) by $z_l$ and sum on $l$, to obtain

$$A(z) = \frac{-8z^3 + 2z - 1}{(2z - 1)^3}$$

after some algebra, so that

$$n_{\mathrm{XX}_l} = [z^l] \frac{-8z^3 + 2z - 1}{(2z - 1)^3},$$

where $[z^l]$ is the coefficient of $z^l$ in the series expansion of the generating function $A(z)$. To extract coefficients, we first decompose $A(z)$ as

$$A(z) = \frac{3}{1 - 2z} - \frac{2}{(1 - 2z)^2} + \frac{1}{(1 - 2z)^3} - 1,$$

which allows us to then find the coefficients as

$$[z^l] \frac{3}{1 - 2z} \overset{(a)}{=} 3 \cdot 2^l, \qquad [z^l] - \frac{2}{(2z - 1)^2} \overset{(b)}{=} -(l + 1)2^{l+1}, \qquad [z^l] \frac{1}{(1 - 2z)^3} \overset{(c)}{=} (l + 1)(l + 2)2^{l-1},$$

where Graham et al. (1994, Table 335) implies (a), (b) is (10) shifted, and (c) is Sloane (1999a) shifted. We omit the last term as it corresponds to $[z^0]$, which we do not need. Now, adding all terms gives

$$3 \cdot 2^l - (l + 1)2^{l+1} + (l + 1)(l + 2)2^{l-1} = 2^{l-1}(l^2 - l + 4),$$

concluding the proof.

## B External results

To proof Proposition 1, we recall McDiarmid's concentration inequality (Vershynin, 2018).

**Theorem 1** (Bounded differences inequality). *Let $X = (X_1, \ldots, X_n)$ be a random vector with independent components. Let $f : \mathbb{R}^n \to \mathbb{R}$ be a measurable function. Assume that the value of $f(x)$ can change by at most $c_i > 0$ under an arbitrary change of a single coordinate of $x = (c_1, \ldots, c_n) \in \mathbb{R}^n$. Then, for any $t > 0$, we have*

$$P\{f(X) - \mathbb{E}f(X) \geq t\} \leq \exp\left(-\frac{2t^2}{\sum_{i=1}^n c_i^2}\right).$$

## C Hyperparameter optimization settings

We collect the hyperparameter choices that we tested in our experiments on real-world classification data (Section 5.2) in Table 3 and refer to the respective original publications for additional information on the parameter settings.

Table 3: Values chosen for the parameter optimization.

| Algorithm | Parameters | Parameter values |
|---|---|---|
| MMDEW | $\alpha$ | $\alpha \in \{0.001, 0.01, 0.1, 0.2\}$ |
| ADWINK | $\delta, k$ | $\delta \in \{0.05, 0.1, 0.2, 0.9, 0.99\}$, $k \in \{0.01, 0.02, 0.05, 0.1, 0.2\}$ |
| D3 | $\omega, \rho, \tau, \mathrm{d}$ | $\omega \in \{100, 200, 500\}$, $\rho \in \{0.1, 0.3, 0.5\}$, $\tau \in \{0.7, 0.8, 0.9\}$, $\mathrm{d} = 1$ |
| IBDD | $m, w$ | $m \in \{10, 20, 50, 100\}$, $w \in \{20, 100, 200, 300\}$ |
| NEWMA | $\omega, \alpha$ | $\omega \in \{20, 50, 100\}$, $\alpha \in \{0.01, 0.02, 0.05, 0.1\}$ |
| Scan $B$ | $B, \omega, \alpha$ | $B \in \{2, 3\}$, $\omega \in \{100, 200, 300\}$, $\alpha \in \{0.01, 0.05\}$ |
| WATCH | $\epsilon, \kappa, \mu, \omega$ | $\epsilon \in \{1, 2, 3\}$, $\kappa \in \{25, 50, 100\}$, $\mu \in \{10, 20, 50, 100, 1000, 2000\}$, $\omega \in \{100, 250, 500, 1000\}$ |

## D Additional experiments

In this section, we collect additional experiments on synthetic data. In Appendix D.1, we align the ARLs of the kernel-based approaches and compare their respective EDD/MTD. In Appendix D.2, we show that our MMD-based approach detects changes in the covariance structure of multivariate data, which aggregated univariate approaches cannot detect reliably.

### D.1 EDD/MTD of kernel-based approaches on synthetic data

The following experiments compare the EDD of the kernel-based online change detection approaches considered in the main text and the related work for a fixed ARL on toy data, extending the experiments of Wei & Xie (2022, Figure 4). We note that all approaches considered in this section compute, for each new observation, a test statistic and compare the statistic to a threshold. If the statistic exceeds the threshold, a change is flagged. In the case of the proposed algorithm, multiple test statistics (one for each possible split) are computed. To allow for a comparison, we select the maximum MMD value across all splits, that is, for MMD, we consider the stopping rule

$$T'' = \inf\left\{t : \max_{s=1,\ldots,l} \mathrm{MMD}\left(\bigcup_{i=s+1}^l \mathrm{X}_i, \bigcup_{i=0}^s \mathrm{X}_i\right) \geq b\right\}, \tag{12}$$

with a fixed $b > 0$, instead of (9) as done in Section 5. Similarly, for MMDEW, we consider (12) but with subsampling applied to the $X_i$-s. The experimental setup is as follows.

To achieve a fixed target ARL $\mathbb{E}_{H_0}T$ for a given stopping time $T$, we run 25 Monte Carlo simulations on 150,000 samples from $\mathbb{P} = \mathcal{N}(\mathbf{0}, \mathbf{I}_d)$ with $d = 20$ and select $b$ as the $1 - 1/(\text{target ARL})$-quantile of the

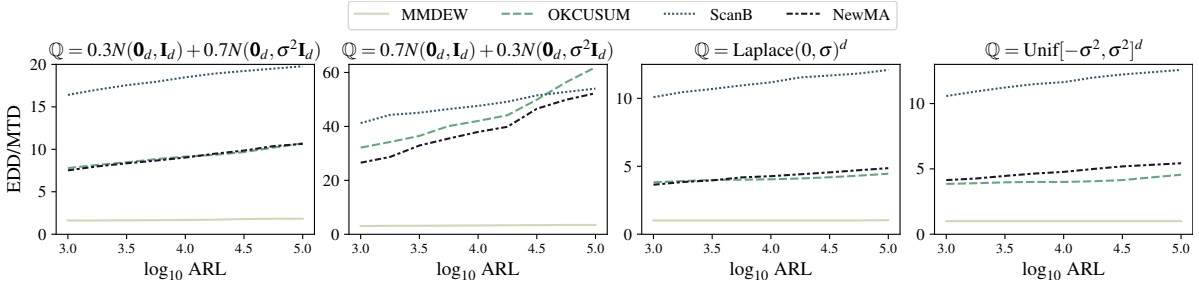

Figure 7: EDD/MTD of kernel-based change detectors with a pre-change distribution of $\mathbb{P} = \mathcal{N}(\mathbf{0}, \mathbf{I}_d)$, $d = 20$, and the indicated post-change distribution ($\sigma = 2$).

collected test statistics as threshold. For online kernel CUSUM, we set its parameters $B_{\max} = 50$ and $N = 15$, matching the settings of Wei & Xie (2022, Figure 4). Similarly, for Scan $B$-statistics and NEWMA, we set $B_0 = 50$; the remaining parameters of NEWMA then follow from the heuristics detailed by the authors (Keriven et al., 2020).

For approximating the EDD of MMDEW and NEWMA for a threshold $b$, we draw 64 and 400 samples from $\mathbb{P}$, respectively, before sampling from $\mathbb{Q}$. Online kernel CUSUM and Scan $B$-statistics each receive 1,000 samples from $\mathbb{P}$ upfront, for computing the variance estimate they require and to use as a reference sample. All approaches use the Gaussian kernel with the bandwidth set by the median heuristic (Garreau et al., 2018).[8]

Figure 7 collects our results, with each subfigure corresponding to a different post-change distribution, of which we sample and process 500 elements to find the first time the test statistics exceeds the threshold. We consider the mean result over 100 repetitions. The results show that OKCUSUM and NEWMA perform similarly across all experiments, with Scan $B$-statistics performing generally worse in three of the cases. MMDEW achieves the lowest EDD throughout. Specifically, on the mixed distribution $\mathbb{Q} = \gamma\mathcal{N}(0, \mathbf{I}_d) + (1 - \gamma)\mathcal{N}(0, \sigma^2\mathbf{I}_d)$ ($\gamma = 0.3$), the EDD of the proposed method is between 1.62 and 1.82. In the more challenging setting of $\gamma = 0.7$, the EDD of MMDEW is between 3.06 and 3.46. Here, OKCUSUM performs second-best, with an EDD of 32.15 for a target ARL of 1,000 and 61.92 for a target ARL of 100,000. On the Laplace and Uniform distributions, the proposed method improves upon the results of OKCUSUM and NEWMA as well, albeit by a smaller margin.

While these experiments show that the proposed method improves upon the state-of-the-art, we note that the experiments require obtaining samples from $\mathbb{P}$, which is rarely feasible in practice. In this case, we recommend setting the threshold of MMDEW by the McDiarmid-based bound (Proposition 1), as we do in the experiments in the main text (Section 5).

### D.2 Comparison with univariate approaches

In this section, we compare the test statistics of the proposed MMDEW, MMD, the Cramer-von-Mises change point model (CvM CPM; Ross & Adams 2012; Ross 2015), and the recent non-parametric Focus (Romano et al., 2023) approach on 20-dimensional multivariate normal data with a mean and correlation shift, respectively. CvM CPM and Focus handle univariate data only. To run each on multivariate data, we run one instance per dimension and consider the means of their test statistics. For MMD and MMDEW, our settings are the same as detailed in Appendix D.1. We set the pre-change distribution to $\mathbb{P} = \mathcal{N}(\mathbf{0}_d, \mathbf{I}_d)$; the respective post-change distributions $\mathbb{Q}$ are indicated in Figure 8. For CPM, which updates all previous test statistics upon observing a new sample, we report the test statistics computed after processing 500 samples from the pre-change distribution and 100 samples from the respective post-change distribution; for all other approaches, we report the test statistic computed upon observing each sample.

---

[8]Note that NEWMA uses random Fourier features Rahimi & Recht (2007) to approximate the kernel.

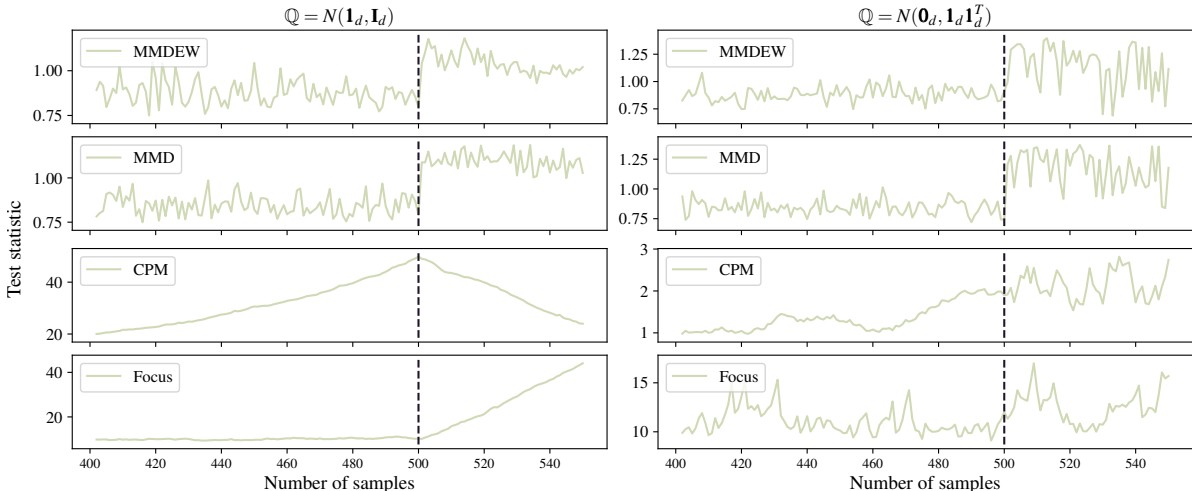

Figure 8: Maximum values of the respective test-statistics (20 repetitions, $d = 20$). A change (indicated by a dashed line) occurs after 500 samples, from $\mathcal{N}(\mathbf{0}_d, \mathbf{I}_d)$ to the distribution indicated on top of the columns, respectively. For the univariate approaches (CPM, Focus), we run one instance per dimension and consider the mean.

Our results are in Figure 8. A change in either the distribution mean (l.h.s.) or the correlation (r.h.s.) lead to an increase of the test statistic of MMDEW and MMD, respectively. Hence, these approaches allow detecting such changes. CPM and Focus correctly identify the change in mean, which is reflected in the univariate marginals they consider. CPM correctly identifies the change point, that is, the maximum value of the test statistic is at 500. When regarding the change in the correlation (the marginals the univariate approaches consider do not change), Focus' test statistic does not reflect the change point. Surprisingly, for CPM, the change in the correlation structure leads to a change in the test-statistic—but the change is identified incorrectly, with the maximum of the test statistic occurring after approximately 530 samples. We conclude that using MMD or the proposed MMDEW is preferable to aggregating univariate change detectors when processing multivariate data, when the changes are not reflected in the marginals.

