# OpenReview forum: "Maximum Mean Discrepancy on Exponential Windows for Online Change Detection"
_TMLR — Accepted by TMLR_

### Review · Reviewer_yR8x · 2024-10-30

**Summary Of Contributions:**

The authors propose a change point detection framework based on MMD. Their method implements temporal windows summarising past observations to achieve lower complexity and memory costs that MMD: $\mathcal{O}(\log^2 t)$ and $\mathcal{O}(\log t)$ respectively.

The proposed method is validated experimentally, against different benchmarks and datasets including standard classification data. In this setup, the proposal performed comparatively well against the considered benchmarks.

**Audience:**

Yes

**Claims And Evidence:**

Yes

**Requested Changes:**

Please see the weaknesses above

**Strengths And Weaknesses:**

**Disclaimer:** This reviewer has worked on CPD in the past but is not up to date with the current state of the art or the relevant challenges of contemporary CPD implementations.  Please take this as an informed, rather than expert, review.

**Strengths**
- The paper is reasonably clear and, up to particular points (see below), successfully delivers the contribution to the reader. There are, however, some formal statements that need to be improved (see below).
- The reference to the relevant previous work seems to be correct (both on CPD and the background on MMD)
- The underlying idea behind the proposed method is valuable and seems to be novel, but it can be better presented (see weaknesses)
- The experiments, in general, seem thorough. Evaluating CPD methods is difficult (due to metrics and datasets), and the authors overcame that challenge by defining meaningful experiments and performance indicators.
- The performance of the proposed method is not always the best, but it doesn't have to be (in order to be a valuable contribution).

**Weaknesses**
- The CPD problem definition in Sec 3 can be explained better. In its current form, one could understand that a different model produces each observation (instead of a _range_ of observations).
- Is there a proof for Prop 1? Is it needed, or would it be redundant given  Gretton et al. (2012, Theorem 8) ?
- Why does Lemma 2 have a proof and Lemma 1 does not?
- Example 1 could be complemented with a diagram
- Proposition 3 is not proven. At this stage, I wonder if these results/claims fall under the Proposition/Lemma category if the don't need to be proven.
- "Figure 1" is not a figure; it's an algorithm

---

> ### Author Response · Authors · 2024-11-01
>
> We thank the reviewer for the very kind review and the helpful suggestions. Below, we address the comments and questions in detail.
>
> **Problem definition.** We agree that our definition allows for every observation to come from a different distribution. While in line with the theory and related work, we will add a sentence that, in practice, a distribution typically generates a sequence of observations.
>
> **Proofs.** We consider the proof of Proposition 1 as one of our core contributions; the proof is in Appendix A.1 "Proof of Proposition 1". The proof of Proposition 3 is in Appendix A.2 "Proof of Proposition 3". The proof of Lemma 1 is stated directly before we state the lemma ("follows from comparing (4) and (5) with (1)"). We will make these connections explicit in the updated version by adding the corresponding references to the statements.
>
> **Figures.** We are happy to include an additional figure (similar to Figure 2) illustrating Example 1 in the revised version of the manuscript and also will rename the algorithm caption to "Algorithm".
>
> We hope this rebuttal answers all the questions and, as per the TMLR instructions, will upload the revised version of the manuscript once all reviews are available.

---

### Review · Reviewer_C2JC · 2024-11-08

**Summary Of Contributions:**

The authors present a novel methodology for performing online non-parametric change-point detection using a two-sample test based on the maximum mean discrepancy (MMD). This approach enables detection of changes between the pre- and post-change distributions in streaming data.

To facilitate efficient, online computation of MMD, the authors introduce an approximation computed over exponentially increasing windows. This allows the method to operate with a logarithmic runtime complexity per operation, specifically:  $O(\log^2(t))$ per update, thereby maintaining a near-constant update rate. This efficiency makes the proposed method suitable for high-speed data streams.

Both simulated and real datasets are present to demonstrate the efficacy of the procedure.

**Audience:**

Yes

**Claims And Evidence:**

No

**Requested Changes:**

1. The paper would benefit from a more comprehensive review of existing procedures in efficient online change-point detection, particularly covering statistical and parametric approaches. A recent litterature review of those approches can be found at:

   - Wang, H., & Xie, Y. (2024). *Sequential change‐point detection: Computation versus statistical performance*. *Wiley Interdisciplinary Reviews: Computational Statistics*, 16(1), e1628.


2. Section 4 could be organized more coherently to enhance readability and ensure that each component is systematically introduced. The following adjustments are recommended:

   a. Notation and Definitions could be improved. The definition of elements such as $B_s$ and associated variables should be clarified to prevent confusion. For example:
     - Explicitly define each window $B_s$ as a triple containing a vector of observations in $\mathbb{R}^s$ and two elements in $\mathbb{R}$ that represent the partial sums and relevant terms.
     - Given the central role of $B$, have the authors considered presenting the entire procedure in terms of $B$?. Currently, there are many mixing terms that could be confusing, such as $X_i$, $x_i$, $x_i^s$, $XX_i$, and $XY_i$. Using something like $B_{s, X}$ instead of XX_s or $B_{s, W}$ instead of X_s.

     - Modify the notation in Equation (3) to indicate that $s$ is a set notation rather than an exponent by placing it in brackets.

   b. Introduce the concept of neighbouring windows earlier and in greater detail. For instance:
     - Equation (7) is somewhat unclear regarding the range of "i." Explain how these ranges operate, especially in cases where split indexing $s$ could be more intuitive.
     - Also, the reviewer is confused. In Figure 2, it seems that, at step 4, the MMD compares elements in $B_0 \cup B_1$ (i.e., points $x_1$ and $x_3$) with $B_2$ (i.e., points $x_1$ and $x_4$), which raises questions about whether overlapping data point x_1 might reduce power?

   c. The pseudocode in the paper could be expanded to ensure it is executable with minimal ambiguity. Specifically:
     - Initialize $B_0$ within the pseudocode.
     - Clarify line 8 for better readability and understanding.
     - Consider evaluating across all possible splits to maximize the likelihood of identifying the change location more accurately, rather than iterating sequentially?

   d. The visual example in Figure 2 is helpful, as it clarifies the process of window merging. Including an expanded, larger-scale visual representation of the window structure could further illustrate the evolution of windows and help with understanding.

3. **Synthetic Data Study**
   The synthetic data study could be used more effectively to highlight both the strengths and potential limitations of the proposed method:

   a. The reviewer is a bit unsure about the setup for Average Run Length (ARL) and Mean Time to Detection (MTD) in the synthetic data analysis. In assessing MTD, it appears that a change occurs only after 512 observations? Since this may not align with real-world streaming contexts where we do not expect to see a change, and changes occur unpredictably, the authors could consider simulating change points later.

   b. the authors could provide an analysis comparing the control and power of the approximated MMD from the proposed procedure with a direct, non-approximate MMD. Though computationally impractical for sequential use, a simple MMD scheme would offer a valuable baseline for assessing the impact of the proposed approximation, particularly on detecting changes appearing at a later time step.

   c. The authors could add a **power comparison** with competitor methods using the synthetic datasets.  The authors could set a consistent alpha level (e.g., $\alpha = 0.05$) across methods with an identical run length (e.g., 10,000 observations). Then, introduce a change at a specified time point (e.g., 5,000) and compare MTD values across a range of replicates for all methods.

   d. The synthetic experiments could also be expanded to:
     - Include simulations with temporal dependency rather than solely i.i.d. data, e.g. a sinusodal process. This would enhance the applicability of the results to real-world settings.
     - Additionally, incorporate non-parametric methods from the statistical literature, such as:
       - Wei, S., & Xie, Y. (2022). *Online kernel CUSUM for change-point detection*. *arXiv preprint arXiv:2211.15070*.
       - Romano, G., Eckley, I. A., & Fearnhead, P. (2023). *A log-linear non-parametric online change-point detection algorithm based on functional pruning*. *IEEE Transactions on Signal Processing*.
       - Ross, G. J. (2015). *Parametric and nonparametric sequential change detection in R: The cpm package*. *Journal of Statistical Software*, 66, 1-20.

**Strengths And Weaknesses:**

The manuscript was an enjoyable read. The authors for prioritizing efficiency, which is essential for online applications, particularly in high-frequency scenarios. The method proposed here has potential as a valuable contribution to the online change-point detection literature due to its reduced runtime complexity and ability to handle continuous data streams effectively.

The reviewer suggests that the paper would benefit from improvements in clarity. Certain areas, in particular those concerning the methodology, could be improved, enhancing readability.

Additionally, it would be beneficial to more clearly address the limitations and shortcomings of the proposed procedure, which are currently underemphasized.

---

> ### Author Response · Authors · 2024-11-20
> **Part 1 (of 2)**
>
> We thank the reviewer for the very in-depth review and the helpful suggestions. Below, we address the comments and questions in detail.
>
> **Comprehensive review.** Wang & Xie (2024) indeed is a very interesting and relevant survey. We now include an extended discussion on existing statistical/parametric approaches in the revised version of the manuscript in Section 2.
>
> **Notational adjustments.** We like the idea of presenting each window $B_s$ as a triplet containing the observations and the summaries and adjusted the notation accordingly. The revised manuscript also emphasizes the role of the superscript $s$ w.r.t. the $x_i^{(s)}$-s. We note that we abstain from making the domain of the $\mathrm{XX}_s$ and $\mathrm{XY}_s$ more explicit, as they vary in length (the first depending on $s$, the last depending on $t$ and $s$), which can complicate parsing these domains as a reader. However, we now include that $\mathrm X_s \in \mathbb R$.
>
> **Range of $i$ in (7).** The $i$ controls which elements are merged, i.e., one fixes a position of a split $s$ and merges all elements coming before the split, merges all elements coming after (and including) the split, and computes MMD between them. Proposition 2 states that our data structure permits the efficient computation of this scheme. We updated the wording in Proposition 2 accordingly, describing the meaning of the split $s$, which clarifies the range of $i$.
>
> **Power reduction.** Between merge operations, no statistical test is performed. Each observation is in at most one of the "samples," and no power reduction due to shared samples occurs.
>
> **Pseudocode.** We updated the pseudocode as suggested to improve clarity. However, we keep the sequential scanning procedure as it corresponds to our implementation and yields the reported results.
>
> **Extended example.**  We happily accomodate the reviewer's suggestion and extended Example 2 and Figure 2, which now features a larger-scale visual representation of the window structure. We further added a figure illustrating Example 1 to the manuscript, as suggested by reviewer yR8x.
>
> **Synthetic data - (a)+(b).** The revised manuscript includes the ARL and MTD results for a change after 1024 samples on different distributions (Section 5.1), addressing (a). We pair these results with a matching set of experiments performed with the quadratic time estimator, addressing (b). For both approaches, we use the McDiarmid-based bound (Proposition 1) to enable a fair comparison. While tighter bounds (e.g., when approximating the null distribution with bootstrapping/permutations) are possible, these induce an immense computational overhead and are infeasible in the streaming case. The experiments show that the quadratic time estimator features a larger ARL, particularly for low values of $\alpha$. While the slower estimator features a lower MTD than the proposed method, the proposed MMDEW still yields competitive results. Further, observing a changepoint after 1024 instead of 512 samples reduces the MTD slightly for both approaches.
>
> **Synthetic data - (c).** We agree that the proposed power comparison is interesting. However, many existing estimators do not come with analytical expressions for their ARL (Table 1) and have many parameters (Table 3). As a fair comparison requires matching the respective ARLs, we consider such experiments out-of-scope of the current work. Our extended ARL results (a)+(b) that the reviewer suggested already support the good ARL/MTD trade-off of the proposed method. Yet, an extended power comparison is interesting future work.
>
> **Non-i.i.d. data - (d).** We agree that handling non-i.i.d. samples is of great interest. However, with our approach relying on the MMD two-sample test, which has an i.i.d. assumption, a direct extension is not directly feasible. We note that the change detection survey Wang & Xie (2024) makes a similar i.i.d. assumption in their Section 2.1 "Problem setup."

---

> > ### Author Response · Authors · 2024-11-20
> > **Part 2 (of 2)**
> >
> > **Comparison to other methods - (d).** The mentioned methods Romano et al. (2023) and Ross (2015) target univariate data, which is different from the setting that we consider. The method by Wei & Xie (2022) "Online kernel CUSUM" (OKCUSUM) handles topological data, as we do, but assumes the availability of a large pool of reference data, which we do not assume. Instead, in our experiments, the algorithms are given a preliminary sample of size 100.
> >
> > Still, as suggested by the reviewer, we experimented with OKCUSUM to compare its ARL and MTD to those of MMDEW.
> > - To compute the respective ARL as given for MMDEW in our Figure 4 (Figure 3 in the original manuscript), we need to find the threshold $b$ (for the stopping rule of OKCUSUM) for a fixed level $\alpha$. Computing such $b$ by Monte Carlo sampling the statistic over the first 100 observations and choosing the $1-\alpha$ quantile yielded an ARL of roughly 100 for small alpha (i.e, $\alpha <= 0.01$) on uniformly distributed samples (dimension $d=5$) with maximal window size $w=20$ and number of windows $N=5$. While larger values of $w, N$ yield slightly better results, these also require a larger sample. In line with the low ARL, the MTD is also very low---but these results are not comparable to those in Figure 3. With these settings of $w$ and $N$, the OKCUSUM statistic also seems to suffer a high variance on uniformly distributed $5$-dimensional data.
> > - The other option is using (12) in Wei & Xie (2022) to approximate $b$ for a fixed ARL. With this methodology, it is unclear how one would obtain an intuitive and fair comparison to the parameter $\alpha$ of our proposed method. Additionally, as in the previous experiment, the choice of $w=20$ and $N=5$ leads to a high variance of the OKCUSUM statistic.
> >
> > For these reasons, we excluded OKCSUM from our experiments but reference it in the related work.
> >
> > We hope this rebuttal answers all the reviewer's questions, and we will upload the revised version of the manuscript soon.

---

> > > ### Comment · Reviewer_C2JC · 2024-12-23
> > > **Revision comment**
> > >
> > > Thank you for addressing my concerns.
> > >
> > > As much as I enjoyed reading the updated version of the paper, I still believe that a power comparison with other approaches over empirical data is within the scope of the paper. This would be useful for potential readers, as they might want to see in practice the scenarios where your approach works best in comparison to competitors, rather than relying on "black box" benchmark tests.
> > >
> > > Regarding the ARL definition, I believe there might have been a slight misunderstanding. I was using the historical definition of *average run length* (see Page, 1954). In statistical literature, this is sometimes referred to as *patience* (Chen et al., 2022) or *expected false detection time* (Yu et al., 2023). It refers to the average stopping time under the hypothesis of no-change for a fixed threshold *b*. In other words, for a fixed threshold, under no change, the ARL tells us the expected number of iterations before the method reports a false positive.
> > >
> > > However, it seems the authors are referring to ARL as the false positive rate up to a fixed run length, which controls something different. I believe this should be clarified, and perhaps aligned with the existing statistical literature.
> > >
> > > ### Tuning Procedures
> > >
> > > Regarding the tuning of the procedures, I understand that matching ARL/FPR could be challenging in practice, especially if one must compute these analytically. However, I do not think tuning, at least the threshold, would be particularly challenging if done via a Monte Carlo study. In fact, in practical applications, where underlying assumptions about theoretically justified bounds may not hold, this approach would be standard. For example, see Section 4.1 of Chen et al. (2022).
> > >
> > > To control the Average Run Length (or patience, as I defined above) for one threshold, this could be achieved fairly easily using the following procedure:
> > >
> > > 1. Generate, say, *K* sequences under the null hypothesis, each of length *N*, where *N* is approximately 10 times the change location.
> > > 2. Apply the approach one wish to tune to each of these sequences and record the maximum value of the statistic for each sequence.
> > > 3. The 1/exp quantile of the distribution of the maximum values of the statistics gives you the empirical threshold.
> > >
> > > Once this empirical threshold is obtained to control the statistics under the null hypothesis, it should be validated against longer sequences (say 3-5 times the average run length) to ensure that the ARL is achieved in practice. This also ensures that the empirical distributions of the run lengths align across all tuned methods, and that the variance of the stopping events is comparable.
> > >
> > > If this is **not** the case, then, as is often the case in non-parametric approaches, the penalty should be tuned based on the false positive rate up to a fixed run length.
> > > This process is very similar to the one described above, except instead of picking the 1/exp quantile, you pick the 1-α quantile (e.g., for those empirical sequences, our approach would produce α * K false positives). This type of comparison aligns the tails of the distributions of stopping events, which allows for a fair comparison of the methods.
> > >
> > > I believe this tuning should be feasible, at least for methods such as NEWMA or the method from Wei & Xie (2022).
> > >
> > > ### Comparing Multivariate and Univariate Approaches
> > >
> > > Additionally, it is common practice to compare multivariate approaches with univariate ones by independently running the method on each covariate and aggregating the independent statistics. For instance,  for each time step, one could sum the independent statistics across each dimension, and then empirically control such a sum as described above. This would provide insights into how the proposed method performs in comparison to simpler, univariate approaches with aggregation, and should be enough to motivate the need for a more sophisticated statistics.
> > >
> > >
> > > ---
> > >
> > >
> > > Chen, Y., Wang, T., & Samworth, R. J. (2022). High-dimensional, multiscale online changepoint detection. *Journal of the Royal Statistical Society Series B: Statistical Methodology*, 84(1), 234-266.
> > >
> > > Yu, Y., Madrid Padilla, O. H., Wang, D., & Rinaldo, A. (2023). A note on online change point detection. *Sequential Analysis*, 42(4), 438-471.
> > >
> > > Page, E. S. (1954). Continuous inspection schemes. *Biometrika*, 41(1/2), 100-115.

---

> > > > ### Author Response · Authors · 2024-12-27
> > > >
> > > > We thank the reviewer for their elaboration and additional remarks, which we comment on in detail below.
> > > >
> > > > __Clarification of ARL.__ Our definition of ARL aligns with the statistical literature referenced by the reviewer; see the definition in Section 5.1 in the second paragraph ("ARL and MTD"). Still, we thank the reviewer for the references, which we will include in the accepted version of the manuscript to clarify this point further. Please note that our threshold directly depends on the level of the statistical test $\alpha$ (which we report); see Proposition 1 and the main algorithm. Accordingly, we indeed fix the threshold.
> > > >
> > > > __Matching of ARL.__ The method the reviewer suggested is very similar to the approach we already took to obtain our results in our reply "Part 2 (of 2)". Accordingly, performing the requested experiments for the stated kernel-based approaches on synthetic data requires only minor changes to our code. We will include the results in the accepted version of the manuscript.
> > > >
> > > > __Comparison to univariate approaches.__ We remark that the current version of the manuscript already compares the proposed method to ADWINK (Faithfull et al., 2019), which runs one instance of the univariate ADWIN algorithm per dimension (i.e., the setting corresponds to the method suggested by the reviewer), on real-world classification data (Section 5.2). Still, we will add a comparison of our approach to other aggregated univariate approaches on multivariate synthetic data, highlighting that our approach can detect changes in the covariance structure even if the marginal distributions stay the same --- a type of change univariate approaches cannot detect.
> > > >
> > > > We hope this rebuttal answers all open questions and will update the manuscript accordingly in the coming days.

---

> > > > > ### Author Response · Authors · 2025-01-03
> > > > >
> > > > > We thank the reviewer for their patience and have updated the manuscript after considering the suggestions.
> > > > >
> > > > > The new experiments are collected in Appendix D. In Appendix D.1, we compare the proposed MMDEW to other kernel-based approaches (NEWMA, online kernel CUSUM, Scan B-statistics) on a superset of the experiments in Figure 4 (Wei & Xie, 2022). After aligning the ARLs, we find that the proposed method achieves a lower EDD than the other approaches on average, further strengthening our results.
> > > > >
> > > > > Appendix D.2 compares our method to the univariate approaches Cramer-von-Mises Change Point Model (CvM CPM; Ross & Adams 2012; Ross 2015) and non-parametric Focus (Romano et al., 2023). The experiments show that MMDEW allows for detecting changes in the covariance structure of a multivariate Gaussian, which the aggregated univariate approaches cannot detect reliably. Changes in the means, reflected in the marginals the univariate approaches consider, are detected by all approaches. As the reviewer suggested, these results highlight the benefit of using multivariate approaches over aggregated univariate approaches for detecting 'challenging' changes.
> > > > >
> > > > > We hope these additional experiments settle all open questions.

---

> > > > > > ### Comment · Action_Editor_BCQF · 2025-01-03
> > > > > >
> > > > > > Hi authors,
> > > > > >
> > > > > > Could you include a precise definition of ARL and MTD? After reading the beginning of Section 5.1, I can more or less formalize the informal definition of MTD (but an actual definition would still be essential to include in the paper), but for ARL, I don't think I follow what you mean exactly, and I can't work that into an actual definition myself. I also agree with the reviewer that the informal definition in the paper (particularly the phrase involving "false positive rate") doesn't seem to match up exactly with what the reviewer had in mind.

---

> > > > > > > ### Author Response · Authors · 2025-01-06
> > > > > > >
> > > > > > > Hello,
> > > > > > >
> > > > > > > we updated the manuscript accordingly and now phrase change detection with MMD as stopping time, which allows us to use the definitions of ARL and EDD common in the literature. We define the stopping time of MMDEW implicitly only, as formalizing the subsampling we perform complicates the notation and, in our opinion, does not contribute to the understanding of the algorithm. Further, we removed the confusing phrase "false positive rate" and thank the AE for pointing this out.
> > > > > > >
> > > > > > > We hope these changes clarify our ARL and EDD/MTD experiments.

---

### Review · Reviewer_t3v5 · 2024-11-17

**Summary Of Contributions:**

The work addresses change detection in data streams using the kernel-based maximum mean discrepancy (MMD) framework, with a primary focus on computational efficiency.

For two sample sets of sizes $m$ and $n$, the time complexity of a single discrepancy test is $O(mn)$. Naively applying this to a data stream of length $t$ requires $O(t)$ discrepancy tests, resulting in a total time complexity of $O(t^3)$. This high computational cost is undesirable in practical applications, and the high frequency of testing may lead to increased false positive rates. The paper's key contribution is a novel algorithm that leverages an efficient online approximation of MMD, reducing the per-observation time complexity to $O(\log^2 t)$. Notably, the proposed algorithm also achieves state-of-the-art F1 scores on 4 out of 5 real-world change detection datasets derived from classification tasks.

The core idea involves segmenting the data stream into exponentially growing windows, with recent observations assigned smaller windows and older observations grouped into larger ones. Relevant statistics are updated online, enabling the computation of MMD test statistics for positions between windows in $O(\log t)$ time.

**Audience:**

Yes

**Broader Impact Concerns:**

There are no concerns.

**Claims And Evidence:**

Yes

**Requested Changes:**

1. The variable $K$ in Equation(2) is undefined. (though I suppose it has the same meaning as the one in Proposition 1?
2. The superscript $s$ in Equations(3-5) is a bit confusing. It will be nice if the authors can provide explicit formulas that map $x_i^s$ to the element $x_j$ in the original input data stream $x_1, \cdots, x_t$.

**Strengths And Weaknesses:**

Strengths. The proposed algorithm achieves the state-of-the-art utility in many real world change-detection datasets. The runtime is also competitive against the other methods.

Weakness. There is little theoretical justification of the proposed algorithm. For example, it would be nice if there were some statistical analysis of the mean time to detection under simple Gaussian data.

---

> ### Author Response · Authors · 2024-11-19
>
> We thank the reviewer for the review and the helpful suggestions. Below, we address the comments and questions in detail.
>
> **Notations.** Indeed, the undefined $K$ matches that of Proposition 1 and we thank the reviewer for the catch. The revised manuscript makes the meaning of $K$ as a bound on the nonnegative kernel explicit. Similarly, we updated the notations and now use $x^{(s)}_i$ ($i=1,\ldots,2^s$) to refer to samples stored in window $B_s$.
>
> **Explicit mapping formula.** Given a stream of data $x_1,\ldots, x_t$, the mapping takes the form $x_i^{(s)} = x_\ell$, with $\ell = \sum_{j=s+1}^{\lfloor\log t\rfloor}2^j[B_j \text{ exists}] +i$, where the bracket is one if the argument is true and zero otherwise (using Iverson's convention). A window $B_j$ exists if the $j$-th right-most digit in the binary expansion of $t$ is $1$. We think this is a valuable addition and include it in the revised manuscript.
>
> We hope this rebuttal answers all the questions and will upload the revised manuscript in the next few days.

---

### Decision · Action_Editor_BCQF · 2025-01-13

**Recommendation:** Accept as is

**Comment:**

As explained above, reviewers and I agree that the paper meets both the "claims and evidence" and "audience" criteria for acceptance in TMLR.

**Audience:**

Changepoint detection is a topic of general interest, and there will be some part of the TMLR audience interested in the result.

**Claims And Evidence:**

After multiple rounds of reviewing+revisions, reviewers are satisfied by the precision of the claims and the theoretical+experimental results substantiating the claims. The authors also clarified certain definitions in the paper upon my request.